# Neurofibromin controls macropinocytosis and phagocytosis in *Dictyostelium*

Gareth Bloomfield[1]*, David Traynor[1], Sophia P Sander[1,2], Douwe M Veltman[1], Justin A Pachebat[3,4], Robert R Kay[1]

[1]MRC Laboratory of Molecular Biology, Cambridge, United Kingdom; [2]Centre for Human Development, Stem Cells and Regeneration, University of Southampton, Southampton, United Kingdom; [3]Department of Plant Sciences, University of Cambridge, Cambridge, United Kingdom; [4]Institute of Biological, Environmental and Rural Sciences, Aberystwyth University, Aberystwyth, United Kingdom

**Abstract** Cells use phagocytosis and macropinocytosis to internalise bulk material, which in phagotrophic organisms supplies the nutrients necessary for growth. Wildtype *Dictyostelium* amoebae feed on bacteria, but for decades laboratory work has relied on axenic mutants that can also grow on liquid media. We used forward genetics to identify the causative gene underlying this phenotype. This gene encodes the RasGAP Neurofibromin (NF1). Loss of NF1 enables axenic growth by increasing fluid uptake. Mutants form outsized macropinosomes which are promoted by greater Ras and PI3K activity at sites of endocytosis. Relatedly, NF1 mutants can ingest larger-than-normal particles using phagocytosis. An NF1 reporter is recruited to nascent macropinosomes, suggesting that NF1 limits their size by locally inhibiting Ras signalling. Our results link NF1 with macropinocytosis and phagocytosis for the first time, and we propose that NF1 evolved in early phagotrophs to spatially modulate Ras activity, thereby constraining and shaping their feeding structures.

*For correspondence: garethb@ mrc-lmb.cam.ac.uk

**Competing interests:** The authors declare that no competing interests exist.

**Reviewing editor**: W James Nelson, Stanford University, United States

## Introduction

Phagotrophic cells feed by performing large-scale endocytosis. A wide range of unicellular eukaryotes grow in this way, suggesting that it is extremely old in evolutionary terms (*Stanier, 1970*; *Cavalier-Smith, 2002*; *Yutin et al., 2009*). Typically phagocytosis is used by these organisms to engulf solid particles (*Metchnikoff, 1892*), and nutrients are then extracted from them by lysosomal degradation (*De Duve and Wattiaux, 1966*). Animal cells and amoebae ingest solid material using F-actin driven projections of their plasma membrane, forming pseudopodia and ultimately cup- or crown-shaped ruffles that enclose adhered particles. These cells can also internalise bulk fluid without the guidance of a particle using a closely related process, macropinocytosis (*Swanson, 2008*).

Phagocytosis and macropinocytosis are controlled using a large set of cytoskeletal and membrane-associated regulators, notably a variety of small G proteins (*Bar-Sagi and Feramisco, 1986*; *Ridley et al., 1992*; *Peters et al., 1995*; *Cox et al., 1997*; *Martínez-Martín et al., 2011*). Oncogenes such as Src and phosphatidylinositide 3′-kinase (PI3K) have also been linked with regulation of these processes (*Araki et al., 1996*; *Veithen et al., 1996*; *Buczynski et al., 1997*; *Amyere et al., 2000*). In amoebae, growth and endocytosis have obvious connections since phagocytosed material supplies essentially all their nutrients; in contrast vertebrates are specialised to digest food extracellularly in the gut, and so links are less apparent. However, large-scale endocytosis is extremely important in immune cells (*Metchnikoff, 1892*; *Norbury et al., 1995*; *Sallusto et al., 1995*), while tumour cells, released from the normal constraints on

**eLife digest** *Dictyostelium* amoebae are microbes that feed on bacteria living in the soil. They are unusual in that the amoebae can survive and grow in a single-celled form, but when food is scarce, many individual cells can gather together to form a simple multicellular organism.

To feed on bacteria, the amoebae use a process called phagocytosis, which starts with the membrane that surrounds the cell growing outwards to completely surround the bacteria. This leads to the bacteria entering the amoeba within a membrane compartment called a vesicle, where they are broken down into small molecules by enzymes. The cells can also take up fluids and dissolved molecules using a similar process called macropinocytosis.

With its short and relatively simple lifestyle, *Dictyostelium* is often used in research to study phagocytosis, cell movement and other processes that are also found in larger organisms. For example, some immune cells in animals use phagocytosis to capture and destroy invading microbes. Most studies using *Dictyostelium* as a model have used amoebae with genetic mutations that allow them to be grown in liquid cultures in the laboratory without needing to feed on bacteria. The mutations allow the 'mutant' amoebae to take up more liquid and dissolved nutrients by macropinocytosis, but it is not known where in the genome these mutations are.

Here, Bloomfield et al. used genome sequencing to reveal that these mutations alter a gene that encodes a protein called Neurofibromin. The experiments show that the loss of Neurofibromin increases the amount of fluid taken up by the amoebae through macropinocytosis, and also enables the amoebae to take up larger-than-normal particles during phagocytosis.

The experiments suggest that Neurofibromin controls both phagocytosis and macropinocytosis by inhibiting the activity of another protein called Ras. Neurofibromin is found in animals and many other organisms so Bloomfield et al. propose that it is an ancient protein that evolved in early single-celled organisms to control the size and shape of their feeding structures.

In humans, mutations in the gene that encodes the Neurofibromin protein can lead to the development of a severe disorder—called Neurofibromatosis type 1—in which tumours form in the nervous system. Given that tumour cells can use phagocytosis and macropinocytosis to gain nutrients as they grow, understanding how this protein works in the *Dictyostelium* amoebae may help to inform future efforts to develop treatments for this human disease.

growth and proliferation, can display pronounced macropinocytotic or phagocytotic uptake (*Lewis, 1937*; *Montcourrier et al., 1994*), and can feed by ingesting extracellular protein (*Commisso et al., 2013*). While there are clear similarities between large-scale endocytosis in animal cells and amoebae, neither the regulatory architecture nor evolutionary contexts are adequately understood.

Phagotrophic microorganisms can be difficult to study in the laboratory because of their requirement for other organisms as food. This can be overcome if cells can be cultured axenically ('a-xenic' indicating the absence of organisms of another species): in some cases, such as the social amoeba *Dictyostelium*, which feeds primarily on bacteria in the wild (*Vuillemin, 1903*), strains were gradually adapted to growth in complex liquid broth, and ultimately in chemically defined media (*Sussman and Sussman, 1967*; *Watts and Ashworth, 1970*; *Franke and Kessin, 1977*; *Watts, 1977*). This process involved the selection of mutants that display increased rates of macropinocytosis (*Watts and Ashworth, 1970*; *Loomis, 1971*; *Hacker et al., 1997*). Two important mutations, *axeA* and *axeB*, were identified by linkage analysis as being necessary for robust axenic growth (*Williams et al., 1974a*, *1974b*), but only the latter is strictly required (*Clarke and Kayman, 1987*). Although these axenic mutant strains have been very widely used for over 40 years, the genetic basis of their growth has remained mysterious, since the mutations could not be precisely mapped. We used a forward genetic approach to identify mutations that promote axenic growth in *Dictyostelium discoideum* using whole genome sequencing. We found that the *Dictyostelium* orthologue of the Ras GTPase activating protein (RasGAP) Neurofibromin (NF1), a tumour suppressor that is mutated in the genetic disorder Neurofibromatosis type 1 (*Xu et al., 1990*), is a key regulator of both macropinocytosis and phagocytosis.

**Table 1**. Mutations in the *axeB* gene in *Dictyostelium discoideum* axenic mutants

| Strain | Mutation | Effect on *Dd* NF1 protein | Position in human NF1 protein |
|---|---|---|---|
| Ax2 | c.-1954_6926delinsCM000150.2: 1390060_1390808 | Deletion to amino acid 2309 | to 2358 |
| AX4 | c.-1954_6926delinsCM000150.2: 1390060_1390808 | Deletion to amino acid 2309 | to 2358 |
| HM557 | c.226_230del | Deletion, frameshift | 66 |
| HM587 | c.1015A > T | Nonsense | 315 |
| HM591 | c.3033_3040del | Deletion, frameshift | ~1060 (in insertion relative to Human) |
| NP73 | c.3508del | Deletion, frameshift | 1228 |
| HM589 | c.4113G > T | K > N | 1423 |
| HM590 | c.4227_4459del | Deletion, frameshift | 1461–1533 |
| HM558 | c.6393_6413inv | DPVVSAIL > EELQKPND | 2182–2189 |
| HM586 | c.6833_7077del | Deletion, frameshift | 2325–2481 |
| HM559 | c.7137_7143del | Deletion, frameshift | 2525–2529 |

Strains are described fully in **Table 2**. Description of changes to the coding sequence of the axeB gene follow the recommendations of the Human Genome Variation Society (**den Dunnen and Antonarakis, 2000**); the effect on the protein sequence is indicated, using the IUPAC one-letter code for amino-acid substitutions. All changes except one are predicted to inactivate the protein either through the introduction of premature stop codons or the substitution of a conserved residue known to be important for function in the human version of the protein. Approximate corresponding locations in the amino-acid sequence of the human orthologue are also indicated.

## Results

### Identification of *axeB*, the major determinant of axenic growth

To generate fresh axenic strains for sequencing, we cultured wildtype *D. discoideum* cells in HL5 growth medium after washing them free of food bacteria. This medium supports the growth of axenic strains such as Ax2 and AX4, but wildtype cells arrest their growth and ultimately die. In order to minimize the number of irrelevant background mutations we avoided mutagenesis and found that spontaneous mutants that are able to grow and proliferate arise frequently among these growth-arrested populations. We selected several independent mutants and sequenced the genomes of three after clonal isolation, along with that of the parental DdB strain, which was chosen because it was also parent to the established axenic laboratory strains (*Bloomfield et al., 2008*).

At first, other than two large duplications that do not correlate with axenicity (*Figure 1—figure supplement 1*), we could only identify one mutation affecting coding sequence in any of these strains relative to their parent, a seven basepair deletion in strain HM559 (*Table 1*). We noted that the reference genome sequence (*Eichinger et al., 2005*), derived from the axenic mutant strain AX4, also differs from its parent DdB in the same gene model (annotated as DDB_G0279251). Further analysis demonstrated that AX4 has lost almost nine kilobases of this region on chromosome 3, resulting in the deletion of most of the coding sequence of a large gene encoding a homologue of the Ras GTPase-activating protein (RasGAP) Neurofibromin (NF1), as well as part of the upstream gene (*Figure 1A*), with a short segment of extraneous sequence inserted. The 7 bp deletion mutation in HM559 lies within the C-terminal region of this NF1 homologue, and we found that another established axenic mutant, Ax2, has exactly the same deletion-insertion mutation as AX4 (*Figure 1—figure supplement 2*; *Table 1*).

Reanalysis of our sequencing data aligned against an amended reference containing this deleted region revealed that both of the other two new mutants also possess mutations in this gene: HM557 has a short frameshifting deletion, while HM558 has undergone an inversion leading to a substitution of eight consecutive amino acids in the predicted protein (*Table 1*). To examine how frequently this gene is mutated in axenic mutants, we amplified and sequenced it from six further strains: five more

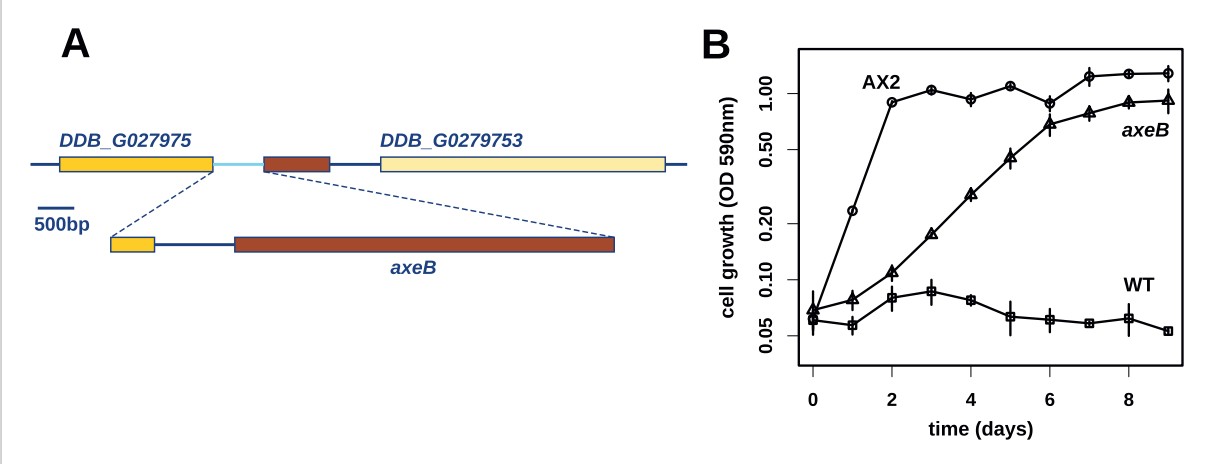

Figure 1. Discovery of the *D. discoideum axeB* locus. (**A**) The region of chromosome 3 spanning the genes DDB_G0279751 and DDB_G0279753 in AX4 genome (top line) contains a conversion mutation in which almost 9 kilobases of sequence (lower line) were lost and replaced by sequence (pale blue) resembling a short region of chromosome 1. The deleted segment contains most of the *D. discoideum* orthologue of NF1, *axeB* (brown). (**B**) NF1 knockout cells can grow in the standard axenic medium, HL5. Amoebae of strains Ax2, DdB (WT), and HM1591 (*axeB*, an engineered NF1 knockout strain in the DdB background; in this and subsequent figures, '*axeB*' refers to this strain), were incubated in tissue culture plates in HL5 medium, and growth measured at indicated timepoints using a crystal-violet binding assay. See also *Figure 1—figure supplements 1, 2*. The AX4 reference genome is at dictyBase (http://dictybase.org).

The following figure supplements are available for figure 1:

**Figure supplement 1**. Two new axenic mutant strains possess overlapping duplications on the same chromosome.

**Figure supplement 2**. Two established axenic mutants possess identical complex mutations affecting the *axeB* gene.

new mutants selected from the same parental DdB strain, and one from the V12 genetic background (strains used in this study are listed in *Table 2*). All possess mutations in the NF1 homologue (*Table 1*): four have frameshifting deletions, one a nonsense mutation, and one has a substitution of a conserved lysine to asparagine.

The ubiquity of mutations in the NF1 gene in the axenic strains tested suggested that they must underlie the phenotype we selected for, and the gene's location on chromosome 3 accords with the mapping of the classically defined *axeB* gene (*Williams et al., 1974a*, *1974b*). To test whether inactivation of NF1 promotes axenic growth, we engineered a deletion at the locus in a wildtype strain, DdB, and found that the resulting mutant is able to grow axenically in HL5 medium (*Figure 1B*). However, it grows more slowly than the established Ax2 strain (*Figure 1B*), and does not grow well in suspension in this medium (see below), confirming earlier findings that additional mutations are necessary to potentiate the basal axenic phenotype (*Williams et al., 1974a*, *1974b*). Together, the identification of mutations in the original axenic strains on chromosome 3 and demonstration that inactivation of the affected gene results in a phenotype closely resembling *axeB* single mutants derived parasexually (*Clarke and Kayman, 1987*) gives adequate reason to believe that we have identified the original causative mutation. We therefore formally retain the name *axeB* for the locus, while naming the encoded protein NF1.

## NF1 is an ancient protein broadly conserved across amoeboid lineages

The *Dictyostelium* NF1 gene encodes a protein with the same domain organisation as the human version, with CRAL/TRIO and PH-like domains at the C-terminal side of the catalytic RasGAP domain (*Figure 2A*). It is also of a similar size, with homology extending across most of the two proteins' lengths (*Figure 2B*). The *D. discoideum* NF1 orthologue is about as similar to the human protein as are those from the basal metazoa and choanoflagellates (*Figure 2C*). NF1 is an ancient protein, conserved considerably beyond the metazoan and fungal lineages in which it has been studied to date, with homologues in a variety of unicellular eukaryotes including the excavates *Naegleria* and *Trichomonas*

**Table 2**. Strains used in this study

| Strain | Parent | Genotype | Reference |
|--------|--------|----------|-----------|
| Ax2 | DdB | axeA2 axeB2 axeC2 | (*Watts and Ashworth, 1970*) |
| AX4 | DdB | axeA1 axeB1 axeC1 | (*Knecht et al., 1986*) |
| DdB | NC4 | Wildtype | (*Bloomfield et al., 2008*, as 'DdB(Wel)') |
| NP73 | V12 | axeB3 | (*Williams, 1976*) |
| HM557 | DdB | axeB(GB1) | This study |
| HM558 | DdB | axeB(GB2) | This study |
| HM559 | DdB | axeB(GB3) | This study |
| HM586 | DdB | axeB(GB4) | This study |
| HM587 | DdB | axeB(GB5) | This study |
| HM589 | DdB | axeB(GB6) | This study |
| HM590 | DdB | axeB(GB7) | This study |
| HM591 | DdB | axeB(GB8) | This study |
| HM1591 | DdB | axeB(GB9) neoR | This study |
| HM1709 | DdB | nfaA(GB1) hygR | This study |
| HM1710 | HM1591 | nfaA(GB1) axeB(GB9) neoR hygR | This study |

The generally accepted genotype of Ax2 and AX4 is given, although the true number of mutations contributing substantially to their fast axenic growth phenotype remains unknown. AX4 derives from another axenic mutant, AX3 (or A3), which was isolated from wildtype cells independently from Ax2 (*Loomis, 1971*). Extant AX3 and AX4 strains share a large inverted duplication on chromosome 2 (*Eichinger et al., 2005*) that is not present in Ax2. However, the mutation in *axeB* in Ax2 and AX4 is identical, suggesting that the extant lines of these strains, along with AX3, had a common ancestor that was axenic. It might not be possible to determine the reason for this discrepancy with the literature; one possibility is that very early in these strains' contemporaneous history one line was contaminated with the other and the slower-growing of the two then lost. In formally numbering alleles we have retrospectively assigned allele number '3' to the *axeB* mutation in the historic strain NP73, but follow recent recommendations (http://dictybase.org/Dicty_Info/nomenclature_guidelines.html) for new strains, and use the same number for gene disruptions using the same knockout construct.

as well as other amoebae (*Figure 2C,D* and *Figure 2—figure supplement 1*; *Carlton et al., 2007*; *Fritz-Laylin et al., 2010*; *Clarke et al., 2013*). RasGAPs are more broadly distributed than NF1, being present in further excavates as well as in certain ciliates, oomycetes, and the foraminiferan *Reticulomyxa* (*Figure 2—figure supplement 2* and *Figure 2—source data 1*; *van Dam et al., 2011*; *Glöckner et al., 2014*). The dictyostelids, *Entamoeba*, *Thecamonas*, and *Naegleria* all possess separate smaller homologues with a similar domain organisation to NF1 but lacking homology outside of the central region; we term these proteins 'MNF' (for 'miniature neurofibromin'). The *D. discoideum* NfaA protein (*Zhang et al., 2008*) falls into this class (*Figure 2A,D* and *Figure 2—figure supplement 1*), and is discussed further below.

The lysine to asparagine substitution occurring in one of our new axenic mutants (see above) has also been found in human cancer and Neurofibromatosis type 1 patients (*Li et al., 1992*), and affects a lysine residue (numbered 1423 in the human polypeptide) that is located on the surface of the GAP domain where it contacts Ras, and that is essential for GAP activity (*Poullet et al., 1994*). This strikingly underscores the homology inferred from sequence analysis, and furthermore suggests that overactivation of Ras subfamily small G proteins causes the axenic mutant phenotype.

## NF1 mutations cause increased fluid uptake

Established axenic mutants have very high rates of macropinocytosis (*Hacker et al., 1997*). To examine fluid uptake in our NF1 knockout mutant, we incubated amoebae with fluorescent dextran and compared them with both the wildtype DdB strain and the established axenic mutant Ax2. When they are harvested directly from bacterial growth plates, NF1 mutants ingest fluid at about the same rate as an established axenic strain, Ax2, and more than four times more rapidly than wildtype cells

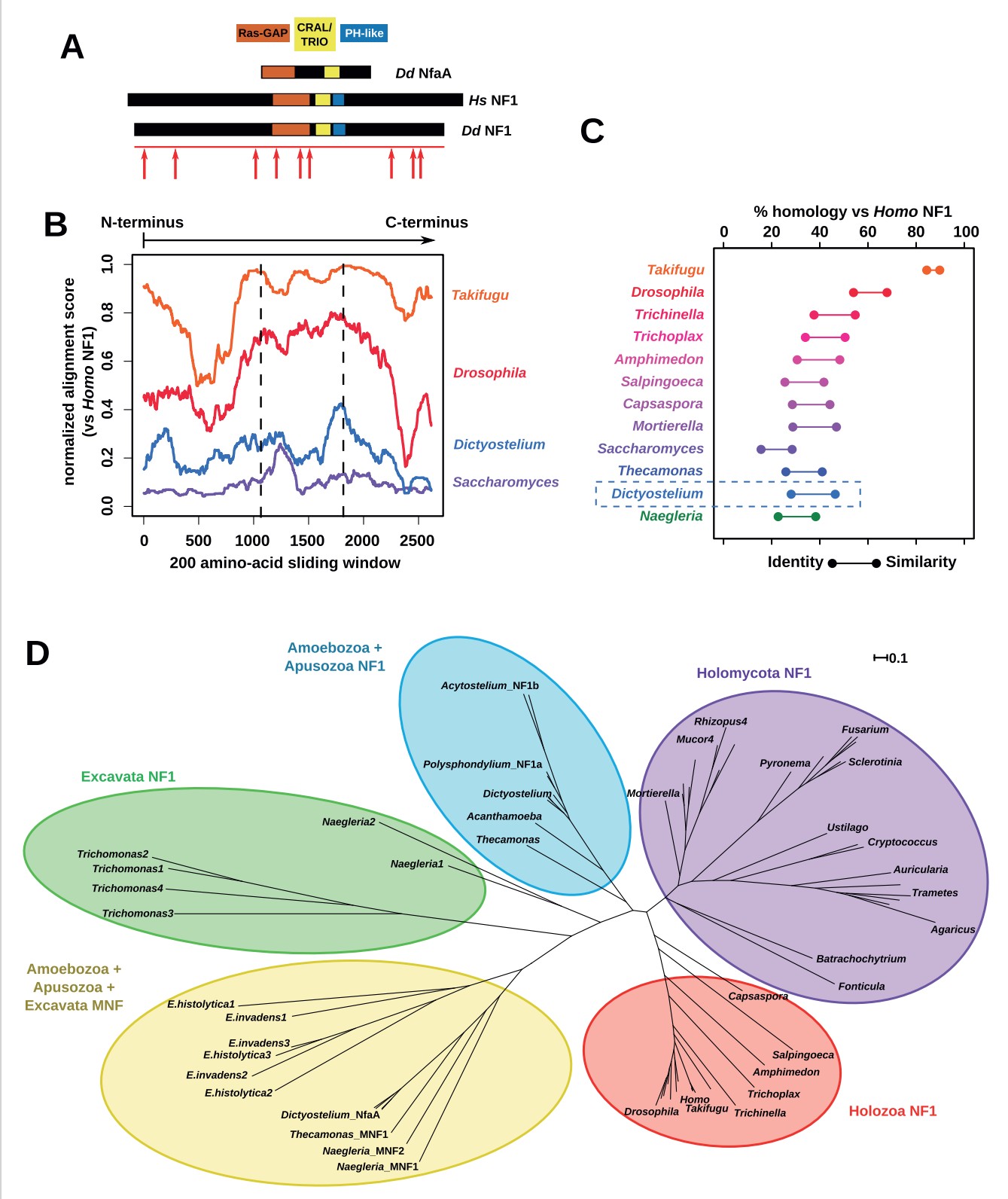

**Figure 2**. NF1 is broadly conserved in a range of amoeboid species as well as animals and fungi. (**A**) NF1 and related proteins have a characteristic domain organisation. The RasGAP domain and adjacent CRAL/TRIO and PH-like domains can be used to identify NF1-like proteins, although the PH-like domain is divergent. Approximate locations of mutations identified in axenic mutants are indicated with arrows; these are described precisely in *Table 1*. (**B**) The *D. discoideum* (*Dd*) NF1 sequence shows homology to the *Homo sapiens* protein along its entire length: the sequence of the *Hs* protein was split into

*Figure 2. Continued*

segments with a sliding window of 200 amino acids, and these globally aligned to the *Dd*, *Takifugu rubripes*, and *Drosophila melanogaster* NF1 orthologues, and the *Saccharomyces cerevisiae* Ira1p sequence. Dashed lines mark the outermost windows containing parts of the central domains. (**C**) NF1 protein sequences from *Takifugu rubripes*, *Drosophila melanogaster*, *Trichinella spiralis*, *Trichoplax adhaerens*, *Salpingoeca rosetta*, *Capsaspora owczarzaki*, *Mortierella verticillata*, *Saccharomyces cerevisiae* (Ira1p), *Dd*, and *Naegleria gruberi* (EFC40840.1) were globally aligned with the *Homo sapiens* NF1 sequence. The bars display the percentage similarity and identity of the protein to the human sequence. (**D**) Phylogram of NF1 and MNF homologues; the Dictyostelium AxeB protein is an NF1 homologue, while homologues of NfaA form the MNF class of RasGAP, defined here. The presence of NF1 and MNF in *Naegleria* and *Thecamonas* as well as amoebozoans indicates that MNF was ancestral and then lost in a common ancestor of the Holozoa and Holomycota after the divergence of apusozoans. The scale shows substitutions/site. See *Figure 2—figure supplement 1* for a version with all species labelled, and also *Figure 2—figure supplements 2* and *Figure 2—source data 1* for illustration of the wider pattern of conservation of RasGAPs.

The following source data and figure supplements are available for figure 2:

**Source data 1**. Examples of RasGAPs and NF1 orthologues in different lineages.

**Figure supplement 1**. Phylogram of NF1 and MNF homologues.

**Figure supplement 2**. The presence of NF1 homologues and other RasGAPs in the three main eukaryotic supergroups.

(*Figure 3A*). After prolonged incubation in HL5 medium without bacteria, Ax2 cells and NF1 mutants increase their uptake further while wildtype cells decrease it, such that after 24 hr mutants take in fluid at a rate more than twenty times higher than the wildtype (*Figure 3A*). Fluid uptake is linear at the earliest times measured with no evidence for rapid recycling of fluid in any of these strains (*Figure 3—figure supplement 1*), in agreement with earlier studies of axenic mutants (*Aubry et al., 1993*; *Padh et al., 1993*). Membrane uptake measured using the accumulation of FM1-43 dye was not increased in NF1 mutants (*Figure 3—figure supplement 2*), consistent with an earlier comparison of Ax2 with wildtype cells (*Aguado-Velasco and Bretscher, 1999*). Since this assay predominantly measures uptake into small vesicles or tubules with a high surface to volume ratio, we conclude that clathrin-dependent and -independent micropinocytotic processes in these cells (*Neuhaus et al., 2002*; *Hirst et al., 2014*), are unaffected by NF1 loss.

Confocal microscopy reveals striking differences between the NF1 knockout mutant and wildtype cells. When bathed in fluorescent dextran, NF1 mutants accumulate large dextran-filled endosomes more prominently than wildtypes (*Figure 3B*), consistent with the fluorimetry of cell populations. NF1 mutants attached to glass coverslips also tend to flatten periodically, a phenotype often observed in established axenic strains such as Ax2 but not in wildtype cells (*Figure 3B*). In four-dimensional timelapse imaging, when freshly harvested from bacteria mutants and wildtype cells can be observed to enclose large macropinosomes after projecting cup- or crown-shaped ruffles, with the mutant performing macropinocytosis in this way four times as frequently as wildtype cells (*Figure 3C*). The NF1 mutation therefore accounts for the increased macropinocytotic fluid uptake of axenic strains.

Despite taking in a similar amount of fluid, the NF1 knockout mutant grows more slowly than Ax2 (see above). We examined whether the mutant processes ingested medium effectively by incubating cells with BODIPY-labelled bovine serum albumin (DQ-BSA), which becomes fluorescent only after lysosomal degradation releases fluorophores that previously quenched each other. Ax2 and the NF1 null strain rapidly and comparably degrade protein after internalization by macropinocytosis and maturation of endosomes, (*Figure 3—figure supplement 3*). Wildtype DdB cells effectively degrade DQ-BSA when taken freshly from bacterial growth but not after overnight incubation in axenic conditions (*Figure 3—figure supplement 3*), again suggesting that they shut down endocytic feeding as part of a starvation response.

## Regulation of Ras activity during macropinocytosis

Given the known function of NF1 in regulating Ras, and the conserved mutation affecting the RasGAP domain in one of our mutants, we examined the involvement of Ras signalling in macropinocytosis, and the specificity of individual RasGAPs in controlling it. First, we deleted the closely related MNF RasGAP, *nfaA*, from wildtype cells, and found that this does not confer the ability to grow axenically (*Figure 4—figure supplement 1*), suggesting that NF1 has specific functions not shared by other

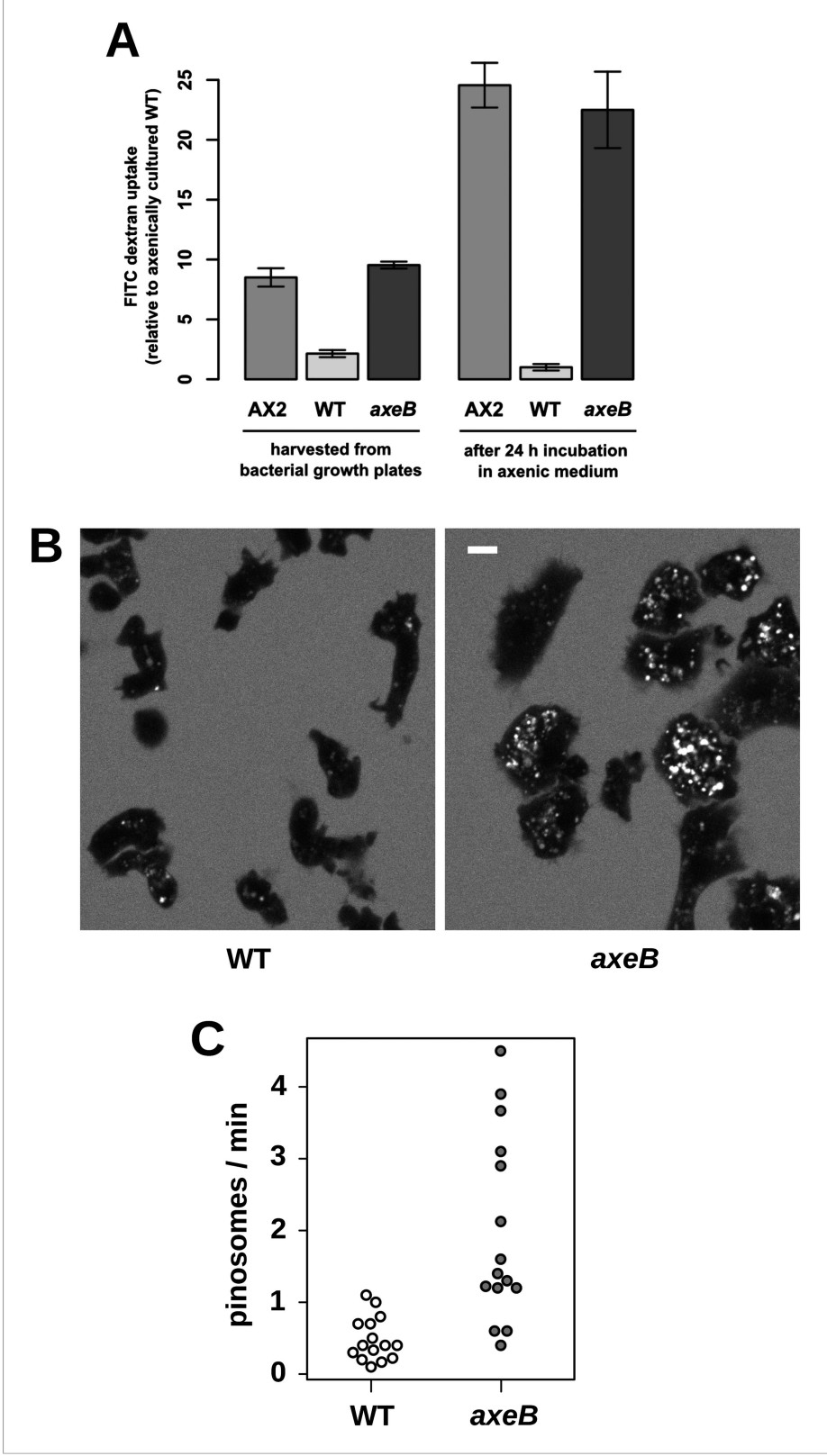

**Figure 3**. NF1 mutants grow axenically in HL5 medium and have increased fluid uptake. (**A**) NF1 knock-out mutants accumulate fluid more quickly than wildtypes. Fluid uptake was measured by shaking cells, either fresh from bacterial growth plates or after 24 hr incubation in axenic medium, with fluorescent dextran in buffer for 1 hr. (**B**) NF1 mutants accumulate fluorescent dextran in large endosomes, and exhibit a flattened phenotype compared
*Figure 3. continued on next page*

*Figure 3. Continued*

to wildtypes. Cells were harvested from bacterial growth plates and incubated in Loflo medium plus TRITC-dextran for 30 min then imaged by confocal microscopy; cells' cytoplasm appears dark since no dextran penetrates it while endosomes are bright as their contents become concentrated. NF1 mutants tend to assume a flattened morphology; since only a single confocal section is shown this will tend to exaggerate the apparent number of endosomes per cell and so these images should not be relied on for comparison of cumulative fluid uptake. (**C**) NF1 knock-out mutants form macropinosomes more frequently than wildtypes, as assessed by confocal imaging. 15 cells of each strain were tracked in total in three independent experiments. Scale = 5 μm. Data points are the means of three independent experiments plus and minus the standard error. See also *Figure 3—figure supplements 1–3*.

The following figure supplements are available for figure 3:

**Figure supplement 1**. No evidence for fast recycling of ingested fluid.

**Figure supplement 2**. In contrast to fluid uptake, membrane uptake is not increased in NF1 mutants.

**Figure supplement 3**. Intracellular degradation of proteins occurs normally in NF1 mutants.

GAPs. This corroborates earlier findings that NfaA has a distinct function regulating pseudopodium formation during chemotaxis (*Zhang et al., 2008*).

We then asked whether inactivation of NF1 results in a global increase in Ras activity. Pulldowns of Ras-GTP using the Raf1 Ras-binding domain (RBD) from growing cells indicated no increase in Ras activity in mutants compared to wildtype cells (*Figure 4—figure supplement 2*). Similarly, confocal microscopy of cells expressing the GFP-RBD reporter revealed no difference between mutant and wildtypes in overall Ras activity estimated by determining the proportion the cell periphery labelled with the RBD (*Figure 4—figure supplement 3*). This is not surprising, since NF1 is only one of twelve putative RasGAPs encoded in the *D. discoideum* genome (not including IQ-GAPs, which generally act as small G protein effectors and scaffolds and do not stimulate Ras GTPase activity [*Shannon, 2012*]).

Having ruled out a global increase in Ras activity resulting from NF1 inactivation, we examined activity at sites of macropinocytosis. In wildtypes, as well as weak localisation at the leading edge of the cell, the GFP-RBD reporter is recruited intensely to small ruffles as they become concave and close into macropinosomes (*Figure 4A*). In NF1 mutants, these specifically macropinocytotic ruffles tend to be larger (*Figure 4A*), 50% of them being greater than 2 μm across upon closure, compared to less than 10% in wildtype cells (*Figure 4B*).

To examine the role of NF1 in regulating macropinocytosis more directly, we expressed full-length constructs of the *D. discoideum* orthologue tagged with GFP. A construct using the native polypeptide sequence reduced axenic growth of the NF1 null mutant considerably (*Figure 4C*), almost completely suppressing macropinosome formation as assessed by confocal microscopy (*Figure 4D*) and displayed an evenly cytosolic distribution (*Figure 4E*). In contrast, a version in which two consecutive arginine residues were mutated in the 'arginine finger' motif required for GAP activity (*Ahmadian et al., 1997*) (NF1-AS) did not retard growth (*Figure 4C*) nor affect macropinosome formation (*Figure 4D*), and localised transiently to membrane ruffles and macropinosomes (*Figure 4E*). This further supports a role for NF1 GAP activity in limiting macropinocytosis. A full length construct expressing the human NF1 protein also adversely affected growth of mutant cells, but a version with the critical arginine residue mutated to alanine reduced growth by a similar amount, suggesting non-specific effects (unpublished data). Since the spontaneous inversion mutation we identified (*Table 1*) affects the C-terminal region of the protein, we also tested whether the core RasGAP-Sec14-PH region of the *D. discoideum* NF1 protein could rescue the NF1 mutant; like the NF-AS version this did not suppress axenic growth (*Figure 4C*), and displayed an evenly cytosolic distribution (*Figure 4E*), suggesting that other regions of the protein are required for its correct localisation. The inactive NF1-AS form of the protein is presumably recruited by strong Ras activity at macropinocytotic crowns that it is then unable to attenuate. This localisation implies that the NF1 protein directly regulates signalling at the plasma membrane during ruffling, and together with the more extensive Ras signalling during macropinocytosis, suggests that this RasGAP regulates a pool of Ras responsible for this feeding process.

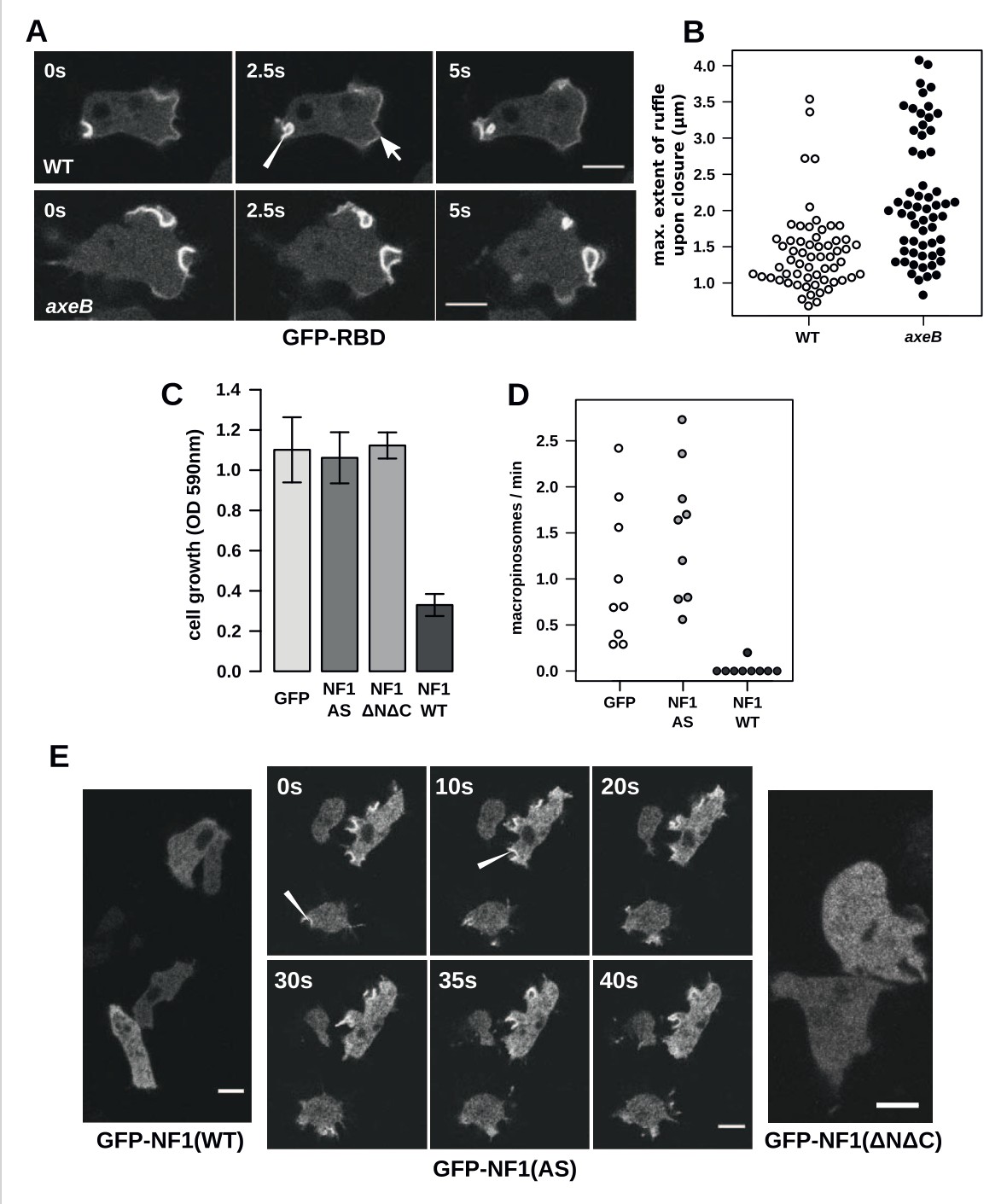

**Figure 4**. NF1 localises to membrane ruffles, its loss potentiates Ras signalling at macropinosomes, and its over-expression represses macropincytosis. (**A**) Ras activity, as reported by GFP-tagged Raf1 Ras-binding domain (GFP-RBD), is exhibited at sites of macropinocytosis (pointer) in wildtype DdB cells as well as at the leading edge (arrow) as the cells move; the distribution of the reporter is qualitatively similar in NF1 knock-out amoebae, but ruffling is more extensive than in wildtypes. (**B**) The Ras-marked membrane ruffles tend to be larger in knock-out mutants prior to closure into pinosomes. Mutant or wildtype GFP-RBD reporter strains were harvested from bacterial growth plates and Ras-marked ruffles were measured across their longest visible axis just after they closed; data are from 60 events for each strain in total from three independent experiments. (**C**) Introduction of N-terminally GFP-tagged Dictyostelium NF1 proteins into *axeB* mutants reduces axenic growth in the case of the wildtype sequence (NF1-WT) but not when two consecutive arginine residues in the protein's 'arginine finger' are mutated to alanine and serine (NF1-AS'), nor when only the central region of the protein encompassing the RasGAP, CRAL-TRIO, and PH-like domains (NF1ΔNΔC) is expressed, when compared to a GFP control. Data are means plus and minus standard error for three independent experiments using the crystal violet assay to assess growth after 7 days incubation in tissue culture plates.
*Figure 4. continued on next page*

*Figure 4. Continued*

(**D**) The active NF1-RR construct almost completely abolishes macropinosome formation when expressed in NF1 mutants, while the inactive NF1-AS form does not inhibit macropinocytosis. Bacterially grown cells were monitored by confocal microscopy as in *Figure 3C*; rates for nine cells of each line from three independent experiments are shown. (**E**) The NF1-AS mutant protein is recruited to membrane ruffles and sites of macropinocytosis (examples indicated by pointers), whereas the wildtype version (NF1-RR) has an even cytoplasmic distribution, as does the truncated NF1ΔNΔC protein. The scale bars represent 5 μm. See also *Figure 4—figure supplements 1–5*.

The following figure supplements are available for figure 4:

**Figure supplement 1**. The axenic growth phenotype is specific to loss of the NF1 RasGAP protein.

**Figure supplement 2**. NF1 mutants do not have an increase in overall Ras activity as assayed using RBD pulldowns.

**Figure supplement 3**. NF1 mutants do not have an increase in overall Ras activity as assessed by confocal microscopy.

**Figure supplement 4**. Localisation of GFP-Ras fusion proteins.

**Figure supplement 5**. Growth phenotypes of Ras expression lines.

Two *Dictyostelium* Ras proteins have been linked to endocytic functions (*Chubb et al., 2000*; *Hoeller et al., 2013*); to examine their involvement in NF1-controlled events we expressed GFP-tagged versions of each in NF1 mutant and wildtype cells. All tested GFP-tagged Ras constructs localised to the plasma membrane, except for dominant negative (S17N mutant) RasG and RasS, expression of which was apparently poorly tolerated in these strains (*Figure 4—figure supplement 4*). None of the *Dictyostelium* Ras expression constructs phenocopied the loss of NF1; constitutively active RasG was deleterious to growth, as was expression of wildtype or constitutively active RasS (*Figure 4—figure supplement 5*), suggesting that improper activation of these isoforms interferes with endocytosis or other Ras-influenced processes leading to detrimental effects on cell growth.

## Downstream signalling events in NF1 mutants

Active Ras at the plasma membrane recruits class 1 phosphoinositide 3′-kinases (PI3Ks; *Rodriguez-Viciana et al., 1997*), allowing the spatially restricted formation of phosphatidylinositol trisphosphate (PIP3) and other inositol phospholipids that occurs during macropinocytosis (*Araki et al., 1996*; *Buczynski et al., 1997*; *Hoeller et al., 2013*). Confirming that the macropinosomes observed in NF1 mutants are mechanistically similar to those previously documented, we find that Ras activity at membrane ruffles in NF1 mutants is accompanied by recruitment of PH-domain reporters that bind the plasmanyl inositides produced by *Dictyostelium* class 1 PI3Ks (*Figure 5A*; *Clark et al., 2014*), as well as by actin polymerisation (*Figure 5B*). PH domains are also prominently recruited during macropinocytosis in wildtype cells; regions of recruitment tend to be larger in mutants reflecting the increased Ras signalling that results from the absence of NF1 (*Figure 5C*). This pattern of Ras activity and PIP3 formation is invariably observed in every instance of macropinocytosis in *Dictyostelium*. The contributions of other Ras effectors remain unclear; for example no increase in ERK activity is observed in NF1 mutants compared to wildtype cells (*Figure 5—figure supplement 1*).

## Wildtype cells are able to grow in complex axenic growth media

The observations described above indicate that wildtype cells perform qualitatively similar macropinocytosis to NF1 mutants, but on a smaller scale. This results in a markedly different outcome when the cells are incubated in HL5 medium: mutants can grow but the wildtype cannot. One possible explanation is that nutrient-uptake below a certain threshold leads to a growth arrest. To test this idea, we asked whether wildtype cells can maintain growth in an enriched axenic medium, as suggested by earlier work (*Sussman and Sussman, 1967*). Wildtype cells incubated in stationary cultures in HL5 supplemented with foetal bovine serum (or bovine serum albumin, data not shown) were able to grow, albeit still much more slowly than NF1 mutants cultured in the same medium (*Figure 6A,B*). The morphology of wildtype cells was not appreciably altered after several days of axenic growth, while NF1 mutants remained consistently more flattened and extensively ruffled than wildtype cells in the same conditions (*Figure 6C*). Wildtype cells were also found to degrade DQ-BSA efficiently after axenic

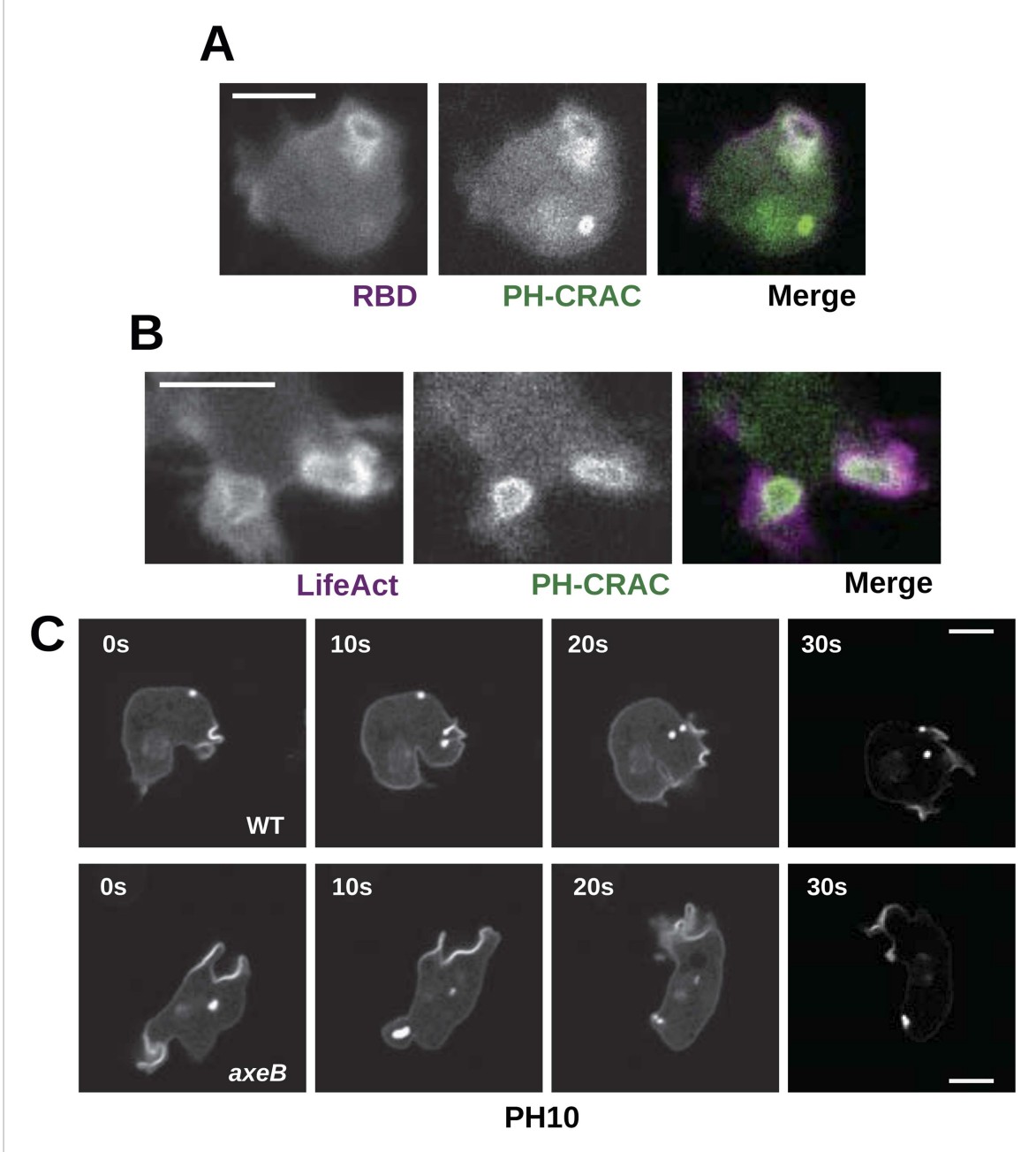

**Figure 5**. Downstream signalling: connections between Ras and PI3K activity during macropinocytosis. (**A**) Ras activity (mCherry-Raf1-RBD reporter, magenta) is accompanied by phosphoinositide 3-kinase activity (PH-CRAC-GFP reporter, green) on macropinosomes in *axeB* mutants; note the green endosome where PI3K products remain but Ras signalling has terminated. (**B**) Actin polymerisation (labelled with mRFP-LifeAct, magenta) occurs around the structures marked by the PH-domain reporter (green). (**C**) PH domains (GFP-PH10) are also recruited to macropinosomes in vegetative wildtype DdB cells; the kinetics of recruitment and retention are similar in *axeB* cells. The scale bars represent 5 µm. See also *Figure 5—figure supplement 1*.

The following figure supplement is available for figure 5:

**Figure supplement 1**. ERK phosphorylation is not increased in NF1 mutants.

growth in the presence of serum (*Figure 6—figure supplement 1*). Serum addition also stimulated the growth of NF1 mutants in shaking suspension (*Figure 6—figure supplement 2*). These findings suggest that the additional nutrients in the richer broth allow these cells to avoid the starvation-triggered growth arrest that can occur in axenic media.

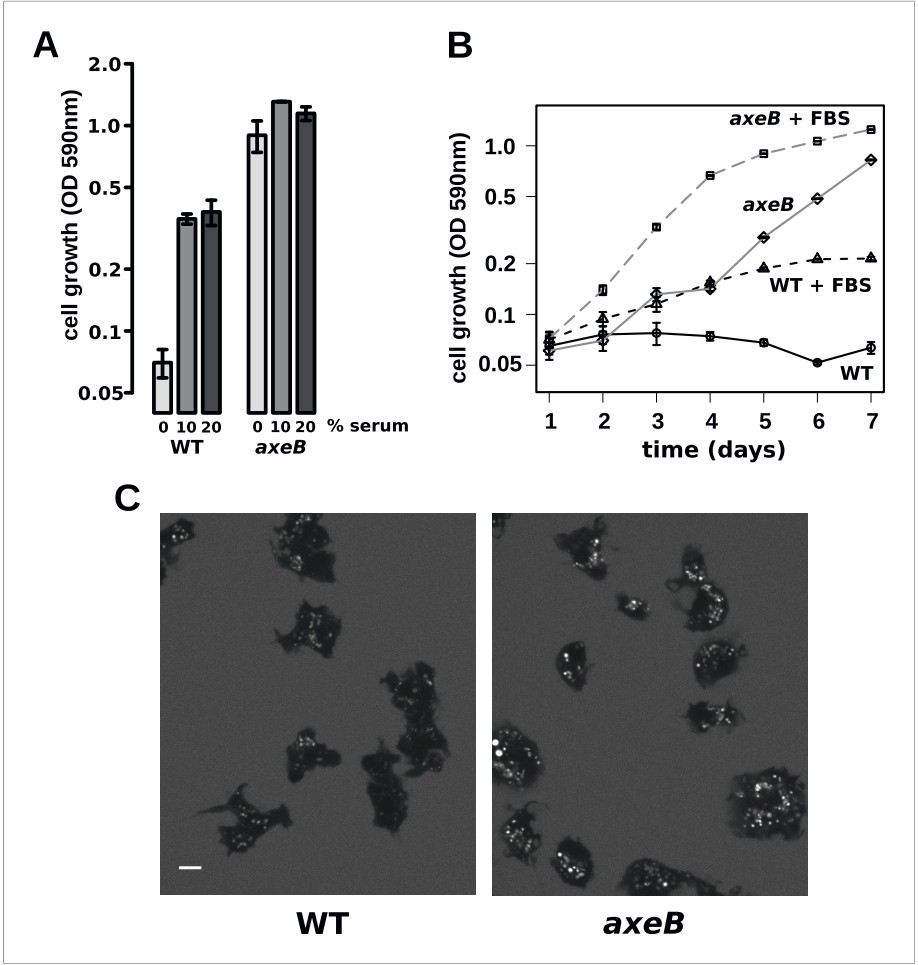

**Figure 6**. Wildtype amoebae can grow axenically in medium supplemented with bovine serum. (**A**) Wildtype (DdB) and NF1 mutant (HM1591) cells were incubated in HL5 medium supplemented with vitamins and microelements without further additions or with 10% or 20% foetal bovine serum (FBS and filter-sterilised HL5 mixed in 1:9 or 1:4 ratios) in 24-well tissue culture dishes at a starting density of $5 \times 10^4$ cells per well. After 7 days growth was measured using the crystal violet assay. FBS stimulated growth of both wildtype and NF1 mutant cells, with mutants having a growth advantage in all axenic conditions. (**B**) Time courses of growth in the presence and absence of 10% FBS in the same conditions as above except that the HL5 medium was dissolved in 10% FBS or in water, then filter-sterilised. Data are means plus and minus standard errors of three (**A**) or four (**B**) independent experiments. (**C**) Wildtype amoebae retain their normal vegetative morphology after growth in serum-supplemented HL5 medium and NF1 mutants are still distinguished by a more flattened appearance. Cells were grown in HL5 plus 10% FBS for 4 days before being washed and placed into Loflo plus 10% FBS in presence of TRITC-dextran. After 30 min, the cells were imaged by confocal microscopy. Scale = 5 μm. See also *Figure 6—figure supplements 1, 2*.
The following figure supplements are available for figure 6:

**Figure supplement 1**. Wildtype cells degrade extracellular protein effectively after growth in rich axenic media.

**Figure supplement 2**. NF1 mutants are able to grow in suspension in rich axenic media.

## NF1 mutants are able to ingest larger-than-normal particles by phagocytosis

Finally, since macropinocytosis and phagocytosis are closely related processes we compared phagocytosis in NF1 mutants and wildtype cells. Mutant and wildtype strains grow well on bacteria (*Figure 7—figure supplements 1, 2*) and take up bacterium-sized polystyrene microspheres (1 μm and 1.8 μm diameter) at very similar rates (*Figure 7A*), although the standard strain Ax2 is marginally

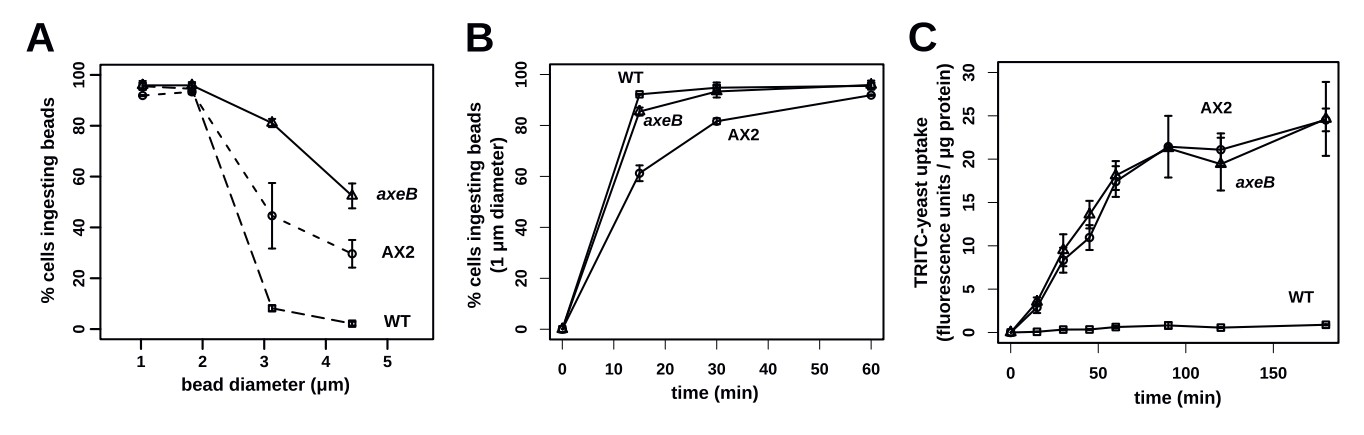

**Figure 7**. NF1 mutants can phagocytose larger particles than wildtypes. (**A**) Axenic mutants ingest small bacterium-sized beads at a similar rate as wildtypes, but wildtype cells are dramatically less efficient at ingesting beads greater than 2 μm in diameter. Cells were harvested from bacterial growth plates, washed, then shaken with fluorescent microspheres of the indicated diameter, then after 1 hr scored for the presence of internalised beads. (**B**) The Ax2 mutant accumulated small 1.0 μm beads more slowly than the wildtype DdB or the *axeB* deletion mutant. (**C**) Axenic mutants can ingest fluorescently labelled budding yeast cells much more easily than wildtype cells. All data are mean ± standard error for three independent experiments. See also *Figure 7—figure supplements 1–3*.

The following figure supplements are available for figure 7:

**Figure supplement 1**. NF1 mutants grow and develop when grown on bacterial lawns.

**Figure supplement 2**. NF1 mutants grow normally when shaken in suspensions of dead bacteria.

**Figure supplement 3**. Phagocytosis is accompanied by Ras and PI3K activity in the same way as in macropinocytosis.

but consistently less effective at internalising smaller beads than the other strains (*Figure 7B*). Against expectation, we found that wildtype cells cannot efficiently ingest yeast or beads greater than 3 μm in diameter (*Figure 7A,C*), whereas NF1 mutant cells can ingest beads larger than 4 μm in diameter (*Figure 7A*) or yeast cells very readily (*Figure 7C*). In line with earlier findings in Ax2 cells (*Clarke et al., 2010*), RBD and PH domain reporters localised to phagosomes as they formed, essentially identically to their behaviour during macropinocytosis (*Figure 7—figure supplement 3*). We conclude that, as well as controlling macropinocytosis, NF1 limits the size of nascent phagosomes, supporting the idea that these large-scale endocytic processes share regulatory as well as structural features. The striking improvement in phagocytosis of larger cells after NF1 deletion also suggests that variation in or loss of this gene can have important ecological and evolutionary consequences by enabling predators to target additional prey species (*Porter, 2011*).

## Discussion

We set out to explain the genetic basis of the axenic growth phenotype of standard laboratory *D. discoideum* strains, which has remained mysterious for decades despite the widespread use of these cells. In freshly selected mutants, we discovered coding sequence mutations only in the *Dictyostelium* orthologue of the tumour suppressor NF1. Importantly, all axenic mutants, across two distinct genetic backgrounds, bear mutations in this gene. While it is possible that mutations in other genes will result in similar phenotypes, it is clear that NF1 mutations must be the most frequent by far that cause axenic growth. The further mutations enabling faster growth in the established axenic strains remain to be identified, and their precise effect is still unclear. Our identification of the *axeB* gene as NF1 will provide a route towards creating new axenic strains from wild isolates, thus giving strains with minimal background mutations.

Vegetative wildtype cells perform macropinocytosis in a qualitatively similar way as axenic mutants but to a lesser extent, and accordingly they can grow axenically when the standard medium is supplemented with bovine serum. NF1 mutants retain a large growth advantage in the more complex

medium, and so will still be selected during prolonged culture. Nevertheless, this protocol should be of use for short-term axenic culture of wildtype strains; standard defined medium supplemented with bovine serum albumin also enables slow growth of wildtype strains (unpublished data) suggesting that fully synthetic defined media should be attainable. This further supports the idea that the important effect of serum is the provision of bulk nutrients, preventing the nitrogen starvation that initiates *Dictyostelium* development (*Marin, 1976*).

One important focus of *Dictyostelium* research is chemotaxis, and key roles for Ras and PIP3 in steering migrating cells have been proposed (*Kay et al., 2008*; *Artemenko et al., 2014*). Although axenic cells chemotax very well to the best-studied chemoattractant, cyclic-AMP, they are much less efficient than wildtype cells in chemotaxis to folic acid, due to interference from large, PIP3-rich, macropinosomes (*Veltman et al., 2014*). It is now apparent that the reason for this poor chemotaxis is likely to be the inactivation of NF1 in axenic cells, leading to Ras and PI3K hyperactivity. Macropinosomes and intense patches of Ras activity and PIP3 also appear in cells chemotaxing to cyclic-AMP (*Parent et al., 1998*) and it will be important to disentangle their contribution to chemotaxis by comparative studies of axenic and wildtype cells.

Several lines of evidence emphasize the importance of Ras in the feeding process used by *Dictyostelium* cells. Previous studies in *Dictyostelium* showed defects in macropinocytosis and phagocytosis after disruption of Ras genes (*Chubb et al., 2000*; *Hoeller et al., 2013*), and Ras activates PI3K during macropinocytosis (*Hoeller et al., 2013*). Further, Ras activity reporters localise strongly to membrane ruffles during macropinocytosis as well as to nascent phagosomes (*Sasaki et al., 2007*; *Clarke et al., 2010*), and we found that the size of these sites of Ras activity is increased in NF1 mutants. The well-studied NF1 orthologues in mammals and yeast are specific to Ras subfamily small G proteins (*Tanaka et al., 1990*; *Zhang et al., 1991*), and one of mutations we found in *Dictyostelium* corresponds exactly to one that abolishes the GAP activity of the human orthologue by altering the Ras-binding interface of the protein (*Poullet et al., 1994*). These data, along with the effect of mutating the 'arginine finger' critical for RasGAP activity, suggest that *Dictyostelium* NF1 is most likely Ras-specific, but this remains to be demonstrated biochemically. The specificity of *Dictyostelium* NF1 towards different Ras isoforms remains unclear. Since the *rasG* and *rasS* null mutants have defective endocytic feeding but are still viable (*Chubb et al., 2000*; *Hoeller et al., 2013*) it is likely that multiple Ras isoforms are involved, possibly including other less well-characterised genes.

The biochemical functions of NF1 beyond its GAP activity remain poorly understood. A module comprising a CRAL/TRIO and a PH-like domain is conserved and likely mediates an interaction with lipids within cells (*D'angelo et al., 2006*), but has not been tied to any function of consequence. Regions of conserved sequence to either side of the relatively well-characterised core of the protein are even more mysterious, but may be important for its function in *Dictyostelium*, perhaps by mediating its dynamic localisation. We find that GFP-tagged full length *Dictyostelium* NF1 localises transiently to macropinocytotic ruffles. This translocation has only been visualised in a mutant form of the protein with the 'arginine finger' motif mutated, because over-expression of the active protein almost completely inhibits macropinocytosis. It will be important to identify structural determinants of this localisation as a route towards a better understanding of the cell-biological function of NF1.

Both macropinosomes and phagosomes are significantly larger in NF1 mutants than in wildtypes, suggesting that NF1 stimulates Ras GTPase activity as endocytic ruffles form and spread, thereby limiting their size (*Figure 8*). The control of NF1 function is not well understood, but our results suggest that it might be locally inactivated during macropinocytosis and phagocytosis. In an intriguing parallel, growth factor treatment of mammalian cells leads to rapid degradation of NF1 by the proteasome (*Cichowski et al., 2003*) and also triggers membrane ruffling and macropinocytosis with similar kinetics (*Brunk et al., 1976*; *Mellström et al., 1983*). However, we could find no obvious fluid-uptake phenotype in NF1 null mouse embryonic fibroblasts, suggesting that NF1 inactivation is not sufficient for stimulation of macropinocytosis in these cells (unpublished data). Given the known involvement of Ras signalling in promoting ruffling and macropinocytosis the possibility remains that NF1 function is conserved in metazoa, but in a context in which Ras activity is more heavily regulated, with additional layers of control not present in amoebae (*Casci et al., 1999*; *Johnson et al., 2005*).

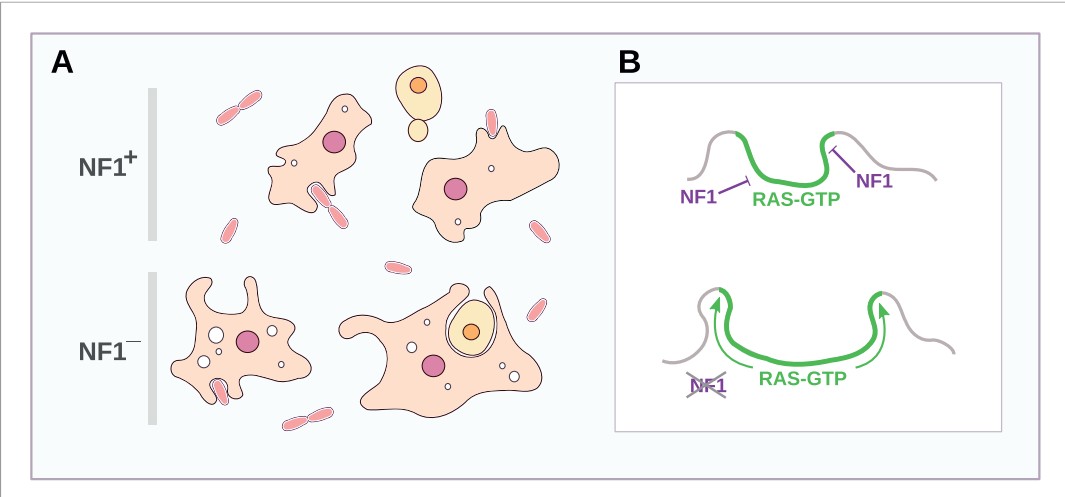

**Figure 8**. Schematic model of NF1 function in Dictyostelium. (**A**) While wildtype NF1+ amoebae ingest bacteria most readily, NF1− cells are also able to ingest larger particles such as yeast cells, and accumulate more fluid in macropinosomes. (**B**) The large concave membrane ruffles formed during phagocytosis and macropinocytosis both are marked by intense Ras signalling (green); NF1 localises dynamically to these regions, stimulating the GTPase activity of Ras proteins there, inactivating them and thereby limiting the expansion and spread of the 'activated' membrane domain.

The set of species possessing NF1 homologues indicates the extreme antiquity of this protein: NF1 is present in the excavates *Naegleria* and *Trichomonas* as well as the amoebae, animals, and fungi. Recent placements of the root of the eukaryotes within the excavate group (*He et al., 2014*; *Derelle et al., 2015*) suggest that NF1 evolved very early in the history of eukaryotes, and may have been present in the last eukaryotic common ancestor (LECA), along with the related protein MNF. The presence of these RasGAPs in organisms that display strikingly similar forms of circular ruffling suggests the hypothesis that NF1 and MNF evolved together to control the movements of the exploratory cell projections used in very early eukaryotes during feeding (*Boschek et al., 1981*; *John et al., 1984*; *Hacker et al., 1997*; *Cavalier-Smith, 2013*).

Macropinocytosis and phagocytosis occur in a well-defined series of stages that are shared between amoebae and vertebrate cells (*Swanson, 2008*). The actin cytoskeleton is used to project membrane ruffles outwards to enclose either a bound particle or extracellular fluid, inositol phospholipids accumulate and are then dephosphorylated in a well-defined sequence (*Dormann et al., 2004*; *Egami et al., 2014*). The role of Ras is less well understood, but has previously been implicated in macropinocytosis and phagocytosis in *Dictyostelium* (*Chubb et al., 2000*; *Hoeller et al., 2013*), and in circular ruffle formation, macropinocytosis, and trogocytosis in vertebrate cells (*Bar-Sagi and Feramisco, 1986*; *Martínez-Martín et al., 2011*; *Welliver and Swanson, 2012*). Our unbiased forward genetic analysis suggests that NF1 has a fundamental role in governing the feeding processes used by amoebae. The mechanistic parallels between large-scale endocytosis in metazoa and amoebae, as well as their shared history, raise the possibility that this function is conserved. The role of Schwann cells, the cell-type of origin of neurofibromas, as non-professional phagocytes during the repair of nerve damage (*Stoll et al., 1989*) is striking in this regard. Although the consequences of NF1 mutation are well understood in humans, its cell-biological function is still not well understood (*Stephen et al., 2014*). Our results, for the first time linking NF1 to macropinocytosis and phagocytosis, may provide an important clue.

## Materials and methods

### *Dictyostelium* cell culture and transformation

*D. discoideum* strains (listed in *Table 2*) were cultivated in association with *Klebsiella pneumoniae* on SM agar plates at 22°C, harvested and prepared for experiments by removing

the bacteria by differential centrifugation in KK2 buffer (16.5 mM $KH_2PO_4$, 3.9 mM $K_2HPO_4$, 2 mM $MgSO_4$). For axenic growth, cells were grown in autoclaved HL5 medium (Formedium, Hunstanton, UK) on tissue-culture treated plastic dishes, or shaken at 180 rpm in 250 ml flasks, in both cases at 22°C. To prepare serum-supplemented medium foetal bovine serum (Hyclone/GE Healthcare, South Logan, Utah) was either added directly to filter sterilised HL5 medium supplemented with vitamins and microelements (HL5VME, Formedium), or first diluted to 10% vol/vol in Milli-Q water then used to dissolve powdered HL5VME before 0.2 μm filter sterilisation. Variable amounts of precipitated material are visible in filtered serum-supplemented HL5VME during incubation with cells at 22°C; these solids may help to stimulate growth (*Watts, 1977*). To measure growth rates on bacteria, cells were shaken in a suspension of heat-killed *Escherichia coli* B/r (at an $OD_{600}$ of 10) in KK2. Cells were transformed according to the method of (*Pang et al., 1999*), except that selection was carried out growing cells on bacteria in order to avoid selection for axenic growth: for the *axeB* gene disruption cells were plated in 2.5 ml KK2 buffer containing a suspension of heat-killed *K. pneumoniae* at an $OD_{600}$ of 5 and 50 μg/ml tetracycline, 100 μg/ml dihydrostreptomycin, and 40 μg/ml G418 in 6-well tissue culture dishes. Once wells became confluent, cells were passaged again in the same conditions to kill non-transformed cells before being cloned on SM agar plates and screened for the disruption by polymerase chain reaction and sequencing. Later transformations, to introduce expression constructs and to disrupt *nfaA*, used an improved selection protocol as follows: *Klebsiella pneumoniae* was grown overnight in a standing bottle of SM/5 broth (Formedium SM broth diluted fivefold in Milli-Q water) to stationary phase, then diluted fourfold in fresh SM/5 broth containing 100 μg/ml dihydrostreptomycin and 60 μg/ml hygromycin, giving a final $OD_{600}$ of approximately 0.1–0.2. A total of 2.5 ml of bacterial suspension was added per well of a 6-well cluster dish, or 10 ml per 100 mm dish. The bacteria and antibiotics were replenished at least every 2 days until amoebae grew to their maximum density (typically 3–4 days after the initiation of selection). Putative *nfaA* disruptants were again cloned on SM agar plates before screening as above.

## Selection of spontaneous axenic mutants

DdB cells were grown on mass culture SM agar plates before being washed free of bacteria and resuspended in HL5 medium. Selection was carried out without mutagenesis by incubation in HL5 medium either under shaking suspension from a starting density of $10^6$ cells per ml (strains HM557, HM558, and HM559) or in 100 mm tissue-culture treated plastic dishes at a starting density of $10^7$ cells per dish (subsequent mutants), at 22°C. Mutants accumulated over the course of 3–5 weeks, and were cloned on SM agar and axenic growth was retested before genome resequencing or targeted sequencing of the *axeB* locus.

## Genome resequencing, read alignment, and variant calling

Genomic DNA was extracted from cells starved overnight that were resuspended in lysis buffer (20 mM Tris-HCl, 5 mM $MgCl_2$, 0.32 M sucrose, 0.02% sodium azide, 1% Triton X-100, pH 7.4) at 4°C, vortexed and incubated at 4°C for 15 min. Nuclei were pelleted at 3000×g for 10 min, resuspended in lysis buffer and pelleted again, before freezing the pellets on dry ice. Proteinase K (100 μl of a 20 mg/ml stock in water) was added, followed immediately by 10 ml digestion buffer (10 mM Tris-HCl, 5 mM EDTA, 0.7% SDS, pH 7.5), and the pellet resuspended by gentle pipetting. The lysate was incubated for 1 hr at 60°C and the DNA finally phenol-chloroform extracted using Phase Lock Gel tubes (5 Prime, Hilden, Germany). Single end Illumina sequencing libraries were constructed according to the manufacturer's instructions. Sequencing was carried out on an Illumina GAII instrument, producing reads of 36 and 45 basepairs across different runs, to a depth of approximately 17–20× after removing of potential PCR and optical duplicates. Reads were aligned against the dictyBase AX4 assembly using Stampy (*Lunter and Goodson, 2010*), and duplicates removed and variants called using samtools and bcftools (*Li et al., 2009*). Candidate variants in each strain were pre-filtered (depth greater than 3, mapping quality greater than 20, SNP quality greater than 20, 'heterozygous' calls excluded) to remove misalignments, then putative variants common to all four resequenced strains, representing real differences between DdB and the reference sequence, were excluded.

## Polypeptide sequence alignments

To display homology across the length of the NF1 protein, the *Homo sapiens* isoform 2 polypeptide sequence (NP_000258.1) was split into segments using a sliding window of 200 residues. These were then aligned using the EMBOSS 'water' local alignment software (*Rice et al., 2000*) to the *D. discoideum*, *Drosophila melanogaster* (AAB58975.1), and *Takifugu rubripes* (AAD15839.1) NF1 orthologues and the *Saccharomyces cerevisiae* Ira1p protein (NP_009698.1). Alignment scores were normalised to the *Hs–Hs* comparison such that the self-comparison gives a value of one, and plotted sequentially along the sequence length for each comparison. To display global similarity and identity percentages, the same *H. sapiens* polypeptide sequence was aligned to the *T. rugripes*, *D. melanogaster*, *Trichinella spiralis* (XP_003376664.1), *Trichoplax adhaerens* (XP_002115170.1), *Salpingoeca rosetta* (EGD75509.1), *Capsaspora owczarzaki* (EFW43762.1), and *Batrachochytrium dendrobatidis* (EGF81694.1) NF1 proteins, *S. cerevisiae* Ira1p and Ira2p (NP_014560.1), and *D. discoideum* NF1 and NfaA (XP_645456.1). The EMBOSS 'needle' software was used to generate global alignments in order that indels count against overall homology.

## Gene disruption and expression constructs

The *axeB* disruption plasmid was constructed by inserting the V18-tn5 cassette into the BamHI and HindIII sites of pBluescript2 KS+, then amplifying 5′ and 3′ flanking regions of the *axeB* gene from strain DdB using the primers KOKpnI (5′-GGTACCAAATGTATACTTGTATAT-GATG-3′) with KOHindIII (5′-AAGCTTGAGCTCTTCACCACCATTAAGT-3′), and KOBamHI (5′-GGATCCATTGGGTAGTTATCGATC-3′), with KOEagI (5′-CGGCCGTGCACAGTCTTTA-GAAAATTTTG-3′), and inserting them either side of it. The *nfaA* disruption plasmid was also based on pBluescript2 KS+ using, between BamHI and XhoI sites, the hygromycin resistance gene driven by the *act14* promoter; and the following primer pairs were used to amplify flanking segments: nfa2kpn (5′-GGTACCTAATGGTGTAACTCAAGTTTTCG-3′) with nfa2xho (5′-CTCGAGTGGTAATGTTT-TATTTGCTGTTG-3′), and nfa2bam (5′-GGATCCAGATATTCATTGTACATCCATCAG-3′) with nfa2spe (5′-ACTAGTATACTTATAAGAAACCTTCTTCAG-3′). Expression constructs used the *act14* or *coaA* promoter to drive the resistance gene and the *act15* promoter to drive the gene fragment of interest, using the vector pDM1005 as backbone, a derivative of previously described extrachromosomal vectors (*Veltman et al., 2009*). The GFP-Raf1-RBD included amino acids 1–134 of the *H. sapiens* Raf1 polypeptide, with a short linker (TTSRT) between GFP and its N-terminal methionine. PH-CRAC-GFP contains residues 1–126 of the *D. discoideum* CRAC protein. GFP-PH10 contains amino acids 1–103 of *D. discoideum* PkgE (*Ruchira et al., 2004*); this fragment is longer than the previously published version, but gives a similar localisation. The GFP-Ras constructs included an RS(GGS)$_4$RS linker between the C-terminus of GFP and N-terminus of each Ras. During the construction of the full length NF1 expression construct, in the same vector backbone including an N-terminal GFP, silent mutations were incorporated at positions 409 and 813 of the *D. discoideum axeB* cDNA sequence to introduce a XhoI and a XmaI site, respectively. The consecutive arginine codons in the 'arginine finger' region were mutated by introducing changes in overlapping PCR primers during cloning. The NF1ΔNΔC construct uses the same vector backbone and tag, and includes amino acids 1189 to 1779, with the wildtype arginine finger motif. Dual-colour experiments used plasmids containing both protein constructs as previously described (*Veltman et al., 2009*).

## Quantitation of *Dictyostelium* cell growth

Since *axeB* knock-out mutants are unable to grow well under shaking suspension, growth was measured by accumulation of crystal-violet staining. Cells were pregrown on bacteria, washed, and plated at a density of $10^5$ cells per well of 24-well tissue culture plates in 1 ml of HL5 medium; for the experiments to test the effect of serum on cell growth the initial density was $5 \times 10^4$ cells per well. At each timepoint the medium was removed, the cells washed once in KK2 buffer, then incubated for 20 min in 0.5 ml of 0.1% crystal violet (in 10% ethanol). Each well was then carefully washed three times with water, then incubated for a further 20 min in 10% acetic acid. After brief agitation, the absorbance of each well was measured at 590 nm. To measure growth on bacteria, cells were grown overnight in heat-killed *E. coli* B/r (OD$_{600}$ of 10 in KK2 buffer) to logarithmic phase then diluted to a density of $5 \times 10^5$ per ml of fresh bacterial suspension; cells were counted using a haemocytometer every 2 hr for 8 hr.

## Fluorescent dextran uptake

Bacterially grown amoebae were assayed either directly or after adaptation in HL5 medium for 24 hr shaken at 180 rpm from a starting density of $2 \times 10^5$ cells per ml. Cells were resuspended at $1 \times 10^7$ per ml in KK2C (KK2 plus 0.1 mM CaCl$_2$) and the assay initiated by adding FITC dextran (average MW 70,000) to 2 mg/ml final with $8 \times 10^5$ cells being removed (in duplicate) for each data point and mixed with 0.75 ml of ice-cold wash buffer (KK2C plus 0.5 mg/ml BSA). The cells were pelleted by centrifugation (20,000×$g$) for 12 s, the supernatant removed and the cells resuspended in 1.5 ml ice-cold wash buffer. The cells were pelleted and washed once more before 1 ml of lysis buffer (0.1 M Tris-Cl pH 8.6, 0.2% Triton X-100) was added. The fluorescent intensity was measured by excitement at 490 nm and emission at 520 nm (PerkinElemer LS50B Luminescence Spectrometer).

## Membrane uptake

Bacterially grown cells were shaken at 180 rpm for 15 min. Then 0.1 ml was added to a stirred fluorimeter cuvette containing 0.9 ml 11 μM FM1-43 (Life Technologies, Paisley, UK) in KK2C and data collected every 1.2 s at an excitation of 470 nm and emission of 570 nm for approximately 5 min using a PerkinElmer LS50B fluorimeter.

## Phagocytosis assays

TRITC labelled yeast was made as described by (*Rivero and Maniak, 2006*) and the assay itself was based on that described in the same paper. Bacterially grown amoebae were resuspended at $2 \times 10^6$ cells/ml in KK2C and the assay initiated by the addition of TRITC labelled yeast cells to approximately $1 \times 10^7$ per ml final. For each data point, $2 \times 10^5$ cells were removed and the uningested fluorescent yeast quenched by the addition of 0.1 ml of trypan blue solution (20 mM sodium citrate, 150 mM NaCl, 2 mg/ml trypan blue). The cell suspension was shaken for 3 min at 2000 rpm (Eppendorf MixMate) and then pelleted by centrifugation (4000×$g$) for 2.5 min. The cell pellet was resuspended in 1.5 ml of KK2C and pelleted as before. Finally, the cell pellet was resuspended in 1 ml of KK2C and the fluorescent intensity measured by excitement at 544 nm and emission at 574 nm (PerkinElemer LS50B Luminescence Spectrometer). For bead uptake experiments Fluoresbrite Bright Blue carboxylate microspheres (Polysciences, Eppelheim, Germany) were used. Bacterially grown cells were resuspended at $10^7$ per ml in KK2C containing 0.2% (wt/vol) BSA (KK2CB) (to reduce non-specific binding) and 20 beads added per amoebae. Uptake was stopped by adding 0.5 ml of the cell suspension to an equal volume of ice-cold KK2CB containing 10 mM NaN$_3$ (KK2CBA). The cells were pelleted at 300×$g$ for 2 min. The pellet was then washed twice more in 1 ml of ice-cold KK2CBA and finally resuspended in 1 ml of ice-cold KK2CBA. 100 μl of this cell suspension was added to 200 μl of KK2CBA containing 40 μg/ml TRITC dextran (as a cell counterstain) in a chamber of a 8-well LabTek chambered coverslip and image stacks taken of several fields of cells for analysis. This procedure removes most non-phagocytosed beads up to 3 μm. For 3.1, and 4.4 μm beads the cells can be directly scored by phase contrast microscopy without counterstain.

## Confocal microscopy

Cells were imaged either directly from growth plates in SM/5 medium or after incubation overnight in Loflo medium, as indicated. To image lysosomal degradation of endocytosed protein, cells were incubated in Loflo medium (Formedium) plus 20 μg/ml DQ green BSA (Life Technologies). Images were acquired using a Zeiss 780 LSM microscope, with laser power and gain set identically for all strains and the brightness and contrast of images adjusted later identically. For DIC images, brightness and contrast was adjusted for visual clarity using ImageJ. To measure the frequency of macropinosome internalization, cells were harvested from mass-inoculation SM agar growth plates, washed three times in Loflo medium, then $1 \times 10^5$ cells plated per chamber of a Lab-Tek II 8-well chambered coverglass (Thermo, Waltham, MA) and allowed to settle for 10–15 min. Within 30 min of removal from bacteria, 0.4 mg/ml FITC- and 2 mg/ml TRITC-dextran were added, and movies recorded taking 5 Z-sections (1 μm apart) every 5 s. Pinosomes were counted if they appeared adjacent to ruffled cell projections and cups, and if they retained FITC fluorescence (FITC is rapidly bleached as endosomes that are acidified). To enable estimation of the rate of uptake, cells were tracked and included in the analysis only if they remained within the field for at least 5 min; these cells

were tracked as long as they remained in the field to at most 10 min. To measure the extent of Ras signalling at the membrane of random growing cells, strains expressing the GFP-Raf1-RBD reporter were harvested during exponential growth in tissue culture dishes containing *K. aerogenes* in SM/5 broth, washed and plated in fresh SM/5 in Lab-Tek chambers as above. Tile scans were acquired of 25 fields, and cells were outlined manually using a custom built MatLab script (*Source code 1*). Normals of 3 pixels long were drawn at equidistant points along the perimeter spaced 2 pixels apart and the highest intensity value along this normal was determined. The patch threshold was set as all membrane values that were more than 3 standard deviations above the mean intensity of the cytosol. Over 100 cells from two independent experiments were analysed for each strain. To measure the extent of Ras signalling during pinocytosis, the same Ras-activity reporter strains were plated in Lab-Tek chambers in SM/5 as above, and movies of a single confocal section through the cells recorded with a 2.5 s interval. The maximum extent of GFP-RBD fluorescence across enclosed ruffles was measured in the first frame after they closed using ImageJ. Images of tagged Ras and NF1 proteins, Raf1-RBD, PH-CRAC, and LifeAct reporters were adjusted for brightness and contrast across the whole image of each channel, and cropped, for clarity.

## Ras pulldowns

Cells were grown in association with *K. aerogenes* on mass-inoculation SM agar plates, washed three times in KK2 buffer, and $2 \times 10^7$ cells resuspended in 10 ml KK2 and shaken at 180 rpm at 22°C for 30 min. The cells were then pelleted at 4°C, and lysed in 1 ml lysis buffer (0.5% Triton X-100, 150 mM NaCl, 40 mM Tris, 20 mM $MgCl_2$, 10% glycerol, 1 mM DTT, 1 tablet per 50 ml Complete EDTA-free protease inhibitors (Roche Lifescience, Burgess Hill, UK), pH 7.4). The lysate was cleared by centrifugation at 13,000×*g* for 10 min at 4°C, then the supernatant added to 33 µl GST-Raf1-RBD conjugated to agarose beads (Millipore, Watford, UK) suspended in lysis buffer, with BSA added to a final concentration of 1 mg/ml. The mixture was then shaken for 30 min at 4°C, before washing twice with lysis buffer by centrifugation at 2000×*g* for 1 min. The bound Ras was released by boiling the washed beads in LDS sample buffer (Life Technologies) for 5 min, before immunoblotting and detection with mouse monoclonal anti-pan-Ras antibody (clone RAS 10, Millipore) and HRP-conjugated goat anti-mouse secondary antibody using standard techniques.

## Detection of phosphorylated ERK

Cells of each strain were harvested from mass inoculation cultures grown in association with *K. pneumoniae* on SM agar, washed, and resuspended in autoclaved HL5 medium at a density of $2 \times 10^6$ cells per ml in a total of 10 ml in 50 ml Erlenmeyer flasks. The flasks were then shaken at 180 rpm and 22°C and aliquots taken at the indicated timepoints. These samples were lysed in LDS sample buffer (Life Technologies) in the presence of protease and phosphatase inhibitors (Roche, as above; Sodium pyrophosphate, sodium orthovanadate, and ß-glycerophosphate), and separated on NuPAGE polyacrylamide gels and blotted onto PVDF according to standard protocols. Blots were blocked in 5% bovine serum albumin in TBS-Tween and activated ERK kinases detected using an antibody raised against phospho-ERK (anti-phospho-p44/p42 MAPK rabbit antibody from Cell Signalling Technology, cat #9101), and HRP-conjugated goat anti-mouse secondary antibody.

## Accession numbers

Sequence data have been deposited in European Nucleotide Archive under the accessions HF565448 (*axeB* genomic sequence) and ERP002043 (whole-genome resequencing reads).

## Acknowledgements

This study was supported by the Medical Research Council (MC_U105115237). SPS was the recipient of 'Leonardo da Vinci' funds as part of the European Commission's Lifelong Learning Programme. Sequencing was carried out at the Cancer Research UK Cambridge Institute, and we are grateful to the Genomics and Bioinformatics facilities there for their assistance. Karen Cichowski generously provided NF1 null and control MEFs. We also wish to thank David Baulcombe, Manu Hegde, Jim Haseloff, Pablo Hollstein, Krys Kelly, and Sean Munro for help and advice.

# Additional information

## Funding

| Funder | Grant reference | Author |
|---|---|---|
| Medical Research Council (MRC) | MC_U105115237 | Robert R Kay |
| European Commission | 'Leonardo da Vinci' programme | Sophia P Sander |

The funders had no role in study design, data collection and interpretation, or the decision to submit the work for publication.

## Author contributions

GB, Initiated the project, Isolated axenic mutants, Prepared sequencing libraries, Analysed sequence data, Engineered knockout mutants and other strains and characterised them, Wrote the paper; DT, Helped to characterise knockout mutants and other strains; SPS, Helped to isolate axenic mutants, to analyse sequencing data, and to characterise knockout mutants and other strains; DMV, Provided unpublished reagents, Helped to characterise knockout mutants and other strains; JAP, Initiated the project, Produced sequencing libraries; RRK, Initiated the project, Isolated axenic mutants, Wrote the paper

# Additional files

## Supplementary files

• Source code 1. MatLab script for quantification of active regions of the cell perimeter.

## Major dataset

The following datasets were generated:

| Author(s) | Year | Dataset title | Dataset ID and/or URL | Database, license, and accessibility information |
|---|---|---|---|---|
| Bloomfield G, Traynor D, Pachebat JA, Kay RR | 2015 | The identification of the Dictyostelium discodeum axeB gene | http://www.ebi.ac.uk/ena/data/view/ERP002043 | Publicly available at the EBI European Nucleotide Archive (ERP002043). |
| Bloomfield G, Traynor D, Pachebat JA, Kay RR | 2015 | Dictyostelium discoideum axeB gene and flanking genes DDB_G0279751 and partial DDB_G0279753, strain DdB | http://www.ebi.ac.uk/ena/data/view/HF565448 | Publicly available at the EBI European Nucleotide Archive (HF565448). |

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
