## [Decision Letter]

Thank you for sending your work entitled “Neurofibromin controls macropinocytosis and phagocytosis in *Dictyostelium*” for consideration at *eLife*. Your article has been favorably evaluated by Randy Schekman (Senior editor), a Reviewing editor, and three reviewers.

The Reviewing editor and the reviewers discussed their comments before we reached this decision, and the Reviewing editor has assembled the following comments to help you prepare a revised submission.

The manuscript describes the identification of the major genetic switch that prevents natural isolates of *Dictyostelium* (i.e. wild type, WT) to survive and proliferate by drinking their environment via macropinocytosis. Mutations that incapacitate the protein encoded by the *axeB* gene, an evolutionarily conserved Ras-GAP that is closely related to the human tumour suppressor NF1, leads to oversized ruffles that renders the *Dictyostelium* strains axenic. Since *axeB* mutants contain exaggerated macropinosomes, as evidenced by the use by various reporters of active Ras signaling, the authors conclude that the main function of NF1 is to restrict Ras activity at the flanks of the nascent macropinosomes (and related the phagosomes).

Major criticisms:

1) It is unclear whether only mutations of NF1 that impact the Ras-GAP activity lead to the axenic phenotype, or are mutations in other domains of this complex protein that do not affect Ras-GAP activity also uncoupling feeding from phagocytosis? Moreover, the data (e.g., Figure 3) show that WT cells do not accumulate fluid phase tracers, and you conclude that the rate of uptake is reduced; is the lack of tracer accumulation in axenic mutants due to slower uptake, faster efflux, lack of concentration, or all three?

2) The substrate specificity of the NF1 Ras-GAP is not explored in vitro (and not so directly in vivo), and the experiments based on expression of various Ras mutants do not replace biochemical evidence. You could make this a strong paper by demonstrating that the *Dictyostelium* NF1 is a true RasGAP acting on one of the Ras proteins involved in macropinocytosis using recombinant proteins in vitro. You have all necessary genes at hand and RasGAP assays can be easily performed with available assays using radioactivity, HPLC, fluorescence or phosphate binding reporters. Alternatively, you could approach this using genetics by investigating the localization of NF1-AS in single or double mutants lacking RasS or RasG. In case the NF1 does not localize in the mutants it is likely that other, as yet uncharacterized Ras proteins are involved.

3) If dissection of the core functions of NF1 in *Dictyostelium* contributes to understanding of the conserved functions of the human NF1 tumor suppressor, then complementation experiments with human NF1 in *Dictyostelium* axenic strains is an easy way to support this claim.

4) The argument in the Discussion section explaining the lack of localization of NF1-WT in *axeB* mutants is difficult to understand as there is no evidence that over-expression of active NF1 leads to loss of macropinosomes. Shouldn't cells lacking macropinosomes die in liquid medium?

[Editors' note: further revisions were requested prior to acceptance, as described below.]

Thank you for resubmitting your work entitled “Neurofibromin controls macropinocytosis and phagocytosis in *Dictyostelium*” for further consideration at *eLife*. Your revised article has been favorably evaluated by Randy Schekman (Senior editor), a member of the Board of Reviewing Editors, and three reviewers. The manuscript has been improved but there are some remaining issues that need to be addressed before acceptance, as outlined below.

All reviewers agreed that you need to make a few changes to the text, but no further experiments are necessary. You were not able to assay RasGAP activity due to the inability to purify a soluble NF1 fragment. It is therefore risky to call the protein a RasGAP at this point without biochemical evidence. A reviewer noted that he/she recalled a similar scenario for the IQGAPs, which based on a similar degree of sequence similarity, were claimed to be RasGAPs. Biochemical assays however later showed that these proteins lack RasGAP activity. The reviewers therefore suggest relativizing this notion. Why not call it a putative RasGAP, or state that it most likely represents a RasGAP?

You were unable to show localization of the NF1-probe in the respective Ras-null mutants. Thus, you employed the pan Ras-probe (Raf1-GBD) to show active Ras in macropinosomes in single Ras mutants. From these results you conclude that RasS and RasG act redundantly (and most likely as substrates for NF1). The reviewers felt that you jumped to the conclusion at this point for the following reasons. We do not know whether the Raf1-GBD probe would localize in the RasS/RasG double Ras-null mutant. If this was not the case, then they are apparently right, but if it still does, then it is likely that additional Ras proteins are involved. They suggest rephrasing this part accordingly.

---

## [Author Response]

*1) It is unclear whether only mutations of NF1 that impact the Ras-GAP activity lead to the axenic phenotype, or are mutations in other domains of this complex protein that do not affect Ras-GAP activity also uncoupling feeding from phagocytosis*?

We know of one mutation outside of the GAP domain that leads to the null phenotype: an inversion mutation that results in eight consecutive amino acid changes towards the C-terminus. This is described in the text and in Table 1. Supporting this, we find that an expression construct truncating the protein before the RasGAP domain after the PH-like domain does not inhibit growth (these data now incorporated into Figure 4). We plan to investigate structure-functional aspects of *Dd* NF1 in a future paper.

*Moreover, the data (e.g.,*
Figure 3*) show that WT cells do not accumulate fluid phase tracers, and you conclude that the rate of uptake is reduced; is the lack of tracer accumulation in axenic mutants due to slower uptake, faster efflux, lack of concentration, or all three*?

We showed in the original manuscript that wildtype (WT) cells form macropinosomes less frequently than NF1 mutants, and that their macropinosomes are smaller, suggesting a primary effect of the NF1 mutation on the rate of fluid uptake. To test this further, we have measured the initial rate of fluid uptake at the shortest times practicable (set by the need for a signal above noise) and find, as expected, that uptake by the wildtype is much slower than NF1 mutants; uptake is linear at these short times arguing against the existence of a fast-recycling compartment (Figure 3—figure supplement 1). Final imaging of WT cells grown axenically in the presence of serum showed that they are able to concentrate ingested fluid effectively (Figure 6—figure supplement 1). Our model remains that WT cells ingest lower volumes of fluid than mutants, and so their growth is arrested by the conventional starvation response unless high levels of nutrients are present (as supplied by HL5 medium with 10% FBS). We have altered the text to make these points clearer.

*2) The substrate specificity of the NF1 Ras-GAP is not explored in vitro (and not so directly in vivo), and the experiments based on expression of various Ras mutants do not replace biochemical evidence. You could make this a strong paper by demonstrating that the* Dictyostelium *NF1 is a true RasGAP acting on one of the Ras proteins involved in macropinocytosis using recombinant proteins in vitro. You have all necessary genes at hand and RasGAP assays can be easily performed with available assays using radioactivity, HPLC, fluorescence or phosphate binding reporters. Alternatively, you could approach this using genetics by investigating the localization of NF1-AS in single or double mutants lacking RasS or RasG. In case the NF1 does not localize in the mutants it is likely that other, as yet uncharacterized Ras proteins are involved.*

We agree that a better knowledge of which Ras proteins are substrates for *Dictyostelium* NF1 would strengthen our paper. While we regard the genetic and homology evidence that NF1 is a RasGAP as compelling, determining which Ras proteins are its substrates has been fraught with difficulty. We had already been attempting to express and purify NF1 fragments for biochemical assays for several months prior to submission of our manuscript, but met with difficulties at every stage, and have so far been unable to purify soluble recombinant protein. Experiments to investigate the localisation of our NF1-AS reporter in mutants lacking different Ras proteins were also unsuccessful because the reporter does not localise in the Ax2 (and AX3) backgrounds where these mutants are available. We note that both RasG and RasS mutants are both partially defective in macropinocytosis and yet the Raf1-RBD reporter still localises to nascent macropinosomes in both strains. This suggests that both RasG and RasS are involved in macropinocytosis (and likely substrates for NF1), but act redundantly.

*3) If dissection of the core functions of NF1 in* Dictyostelium *contributes to understanding of the conserved functions of the human NF1 tumor suppressor, then complementation experiments with human NF1 in* Dictyostelium *axenic strains is an easy way to support this claim.*

We agree that complementation of the *Dictyostelium* NF1 mutant with the human protein is a most desirable experiment. Although we have had success with the highly conserved SBDS protein in the past, the *Dictyostelium* NF1 is less well conserved (although the clear orthologue of the human protein) and we doubted that NF1 protein of one species would interact sufficiently well with the orthologous binding partners to achieve full complementation. Nevertheless, we engineered a construct to express GFP-tagged full length human NF1, with the codons optimized for *Dictyostelium*. Expression of this construct did indeed strongly reduce axenic growth of the NF1 mutant; unfortunately, so did a version in which the catalytic arginine residue of the GAP domain was changed to alanine. This experiment will be followed up in various ways but in the meantime we feel it is too difficult to interpret to merit inclusion in the body of the paper. Despite the evolutionary distance between *Dictyostelium* and humans, we firmly expect that determination of the basal function of NF1 in cellular contexts closely resembling those in which it first evolved (as we argue from our phylogenetic analysis) will deepen our understanding of its function in metazoan cells.

*4) The argument in the Discussion section explaining the lack of localization of NF1-WT in axeB mutants is difficult to understand as there is no evidence that over-expression of active NF1 leads to loss of macropinosomes. Shouldn't cells lacking macropinosomes die in liquid medium*?

We show in Figure 4 that expression of the wildtype NF1 protein in the *axeB* mutant strongly reduces growth in axenic medium. We have now quantified macropinosome uptake to substantiate this aspect of the phenotype: uptake of macropinosomes is almost completely abolished, and this reporter always appears to have an even cytosolic distribution. We have clarified the text accordingly.

[Editors' note: further revisions were requested prior to acceptance, as described below.]

*All reviewers agreed that you need to make a few changes to the text, but no further experiments are necessary. You were not able to assay RasGAP activity due to the inability to purify a soluble NF1 fragment. It is therefore risky to call the protein a RasGAP at this point without biochemical evidence. A reviewer noted that he/she recalled a similar scenario for the IQGAPs, which based on a similar degree of sequence similarity, were claimed to be RasGAPs. Biochemical assays however later showed that these proteins lack RasGAP activity. The reviewers therefore suggest relativizing this notion. Why not call it a putative RasGAP, or state that it most likely represents a RasGAP*?

*You were unable to show localization of the NF1-probe in the respective Ras-null mutants. Thus, you employed the pan Ras-probe (Raf1-GBD) to show active Ras in macropinosomes in single Ras mutants. From these results you conclude that RasS and RasG act redundantly (and most likely as substrates for NF1). The reviewers felt that you jumped to the conclusion at this point for the following reasons. We do not know whether the Raf1-GBD probe would localize in the RasS/RasG double Ras-null mutant. If this was not the case, then they are apparently right, but if it still does, then it is likely that additional Ras proteins are involved. They suggest rephrasing this part accordingly*.

We agree that the additional qualifications to our conclusions regarding the potential RasGAP activity of *Dd* NF1 and its substrates are justified given the remaining uncertainties. We have addressed this by amending the unfortunate construction of the final sentence of the Introduction, and further clarifying the paragraph in the Discussion that outlines the evidence for the specific involvement of Ras subfamily proteins in the phenomena of interest. I should note also that we have reverted to the more usual 'PI3K' as the abbreviation for 'phosphoinositide 3'-kinase' at the insistence of the senior author.